# PaLI-3 Vision Language Models: Smaller, Faster, Stronger

## Abstract

This paper presents PaLI-3, a smaller, faster and stronger vision language model (VLM) that compares favorably to similar models that are 10x larger. As part of arriving at this strong performance, we compare Vision Transformer (ViT) models pretrained using classification objectives to contrastively pretrained ones (SigLIP). We find that, while slightly underperforming on standard image classification benchmarks, SigLIP-based PaLI shows superior performance across various multimodal benchmarks, especially on localization and text understanding. The SigLIP encoder we use is a scaled-up version using 2 billion parameters, and achieves a new state-of-the-art on multilingual cross-modal retrieval. We consider that PaLI-3, at only 5B parameters, rekindles research on fundamental pieces of complex VLMs, and could fuel a new generation of scaled-up models.

## 1 Introduction

The scaling of vision-language models (VLM) to tens and even hundreds of billions of parameters (Chen et al., 2023b; Alayrac et al., 2022; Chen et al., 2023a; Driess et al., 2023) has shown ever-increasing performance. Meanwhile, models at a smaller scale remain critical, as they are more practical to train and serve, more environmentally-friendly, and support faster research cycles for model design.

In the spirit of focusing on smaller-scale modeling, we present PaLI-3, the third-generation family of PaLI (Chen et al., 2023b) models. Using a pretrained backbone with only 5B total parameters, we refine the training recipe and achieve competitive and new state-of-the-art (SOTA) results on various VLM benchmarks. Our new recipe has three main components: contrastive pretraining of image encoder on web-scale image-text data (Zhai et al., 2023), an improved dataset mixture for PaLI multimodal training, and training at higher resolutions.

PaLI-3 achieves new SOTA results on tasks that require visually-situated text understanding and object localization, including eight visually-situated text understanding tasks and the referring expression segmentation task on RefCOCO (Yu et al., 2016), along with strong performance on a wide range of classical vision tasks. As part of this work, we also introduce a SOTA multilingual contrastive vision model scaled to 2B parameters, obtained using the recently-introduced SigLIP recipe (Zhai et al., 2023). Additionally, we perform focused ablation studies to compare the classification pretrained Vision Transformer (ViT) backbones (Dosovitskiy et al., 2021) with contrastively pretrained ones (SigLIP). This further confirms the viability of pretraining visual encoders on noisy web-scale image-text data, as a preferable alternative to training on classification-style data.

Our contributions are summarized as follows:

1. We compare classification pretrained ViT models (Dosovitskiy et al., 2021) to contrastively pretrained SigLIP models using the PaLI framework (Chen et al., 2023b). We find that the contrastively pretrained models work significantly better for visually-situated text understanding tasks and localization tasks.

2. Our model achieves SOTA performance on 10+ diverse vision-language benchmarks while being 10x smaller in size compared to the current SOTA model Chen et al. (2023a). For understanding visually-situated text, the improvement is by a particularly large margin.

3. Despite not pretraining on any video data, our model achieves new SOTA on several video QA benchmarks, indicative of powerful generalization abilities.

4. We introduce the 2 billion parameter (ViT-G) multilingual SigLIP model trained on We-bLI (Chen et al., 2023b), which sets a new state-of-the-art on the multilingual cross-modal retrieval benchmark Thapliyal et al. (2022) across 36 languages.

## 2 RELATED WORK

Recent large vision language models (VLMs) use pretrained image encoders as part of the larger model, some pretrain it with supervised classification (PaLI (Chen et al., 2023b), PaLI-X (Chen et al., 2023a), Flamingo (Alayrac et al., 2022), PaLM-E (Driess et al., 2023)), some use pretrained CLIP encoders (BLIPv2 (Li et al., 2023), CrossTVR (Dai et al., 2023), ChatBridge (Zhao et al., 2023)) and some with custom multimodal pretraining (BEiT3 (Wang et al., 2022b), CoCa (Yu et al., 2022), SimVLM (Wang et al., 2021b)). In this paper we compare two dominant ways to pretrain image encoders using the PaLI framework: classification pretraining using large weakly labeled datasets (JFT, as in Kolesnikov et al., 2020; Zhai et al., 2022a; Dehghani et al., 2023) and contrastive pretraining on web-scale noisy data (WebLI, as in Zhai et al., 2023).

A recent finding spanning across PaLI (Chen et al., 2023b) and PaLI-X (Chen et al., 2023a) is that scaling up the classification pretrained image encoder seems more promising than was previously believed (Alayrac et al., 2022). Specifically, while classic image-only benchmarks such as ImageNet seem to indicate saturating performance from scaling pretraining of image-only models (Zhai et al., 2022a), PaLI shows that by scaling up the vision encoder from ViT-G (2B) to ViT-e (4B), the improvements on VL tasks are more noticeable than on ImageNet. PaLI-X further scaled up both the vision and language components, showing that these larger image encoders keep bringing benefit when plugged into large VLMs. This finding suggests that there is more to be found regarding the pretraining of image encoders in the context of VLMs, which may lead to different conclusions when looking at VLM tasks as compared to of "pure" vision tasks. In this paper, we dive into the impact of the image encoder for VLMs, by directly comparing classification pretrained vision models to contrastively pretrained ones, and reveal that the latter are vastly superior on a variety of tasks, especially localization and visually-situated text understanding.

One can split multimodal understanding capabilities into largely two categories: natural scene understanding (captioning, VQA, object detection/localization), and visually-situated text understanding (document and infographics QA). These groups of tasks require different granularity of understanding, and previous VLMs have largely focused on one type of tasks, leading to their training recipes being attuned to that type of tasks. For example PaLI-17B (Chen et al., 2023b) and Pix2struct (Lee et al., 2022) showed strong performance only on one of the two categories, respectively. The recent PaLI-X (Chen et al., 2023a) achieves SOTA performance on both categories, based on an improved OCR-related training recipe, and a significantly larger 55B parameter model. In this work, we combine the advantage of contrastively-pretrained ViT and a further improved and balanced training recipe into PaLI-3, and demonstrate that SOTA level performance on both the above categories of multimodal understanding is achievable even at 5B parameter scale.

## 3 MODEL

### 3.1 ARCHITECTURE

On a high level, the architecture follows Chen et al. (2023b;a): a ViT encodes the image into tokens which, together with text input (the question, prompt, instruction), are passed to an encoder-decoder transformer (Vaswani et al., 2017) that generates a text output.

**Visual component** The vision backbone of PaLI-3 is initialized from a contrastively pretrained ViT-G/14[1] (Zhai et al., 2022a) model (approx. 2B parameters) using the SigLIP (Zhai et al., 2023) training recipe. In short, an image embedding ViT-G/14 and a text embedding transformer are

---

[1]The embedding dimension was changed from 1664 to 1536 for better hardware utilization.

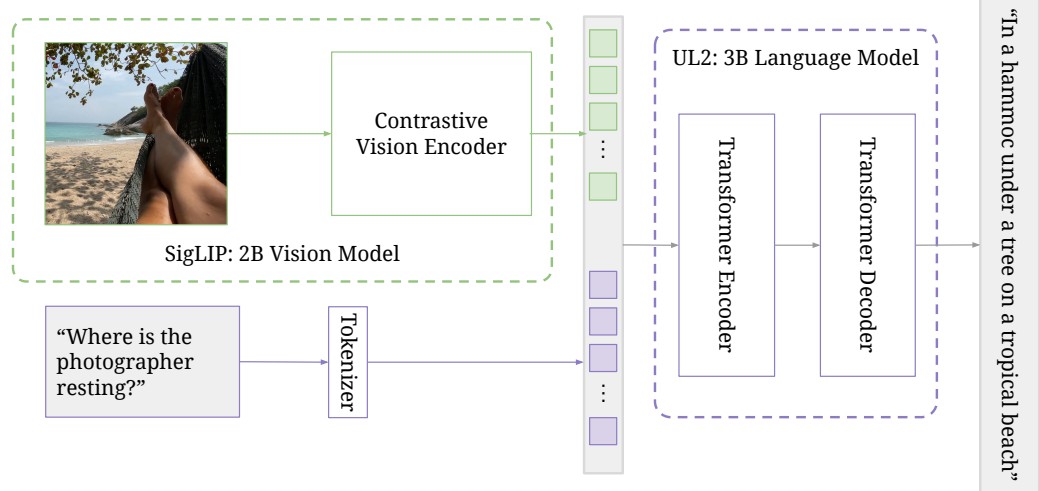

Figure 1: Overview of the PaLI-3 (5B) model: images are encoded into visual tokens individually by the contrastively pretrained 2B SigLIP vision model. Along with a query, these visual tokens are passed to an 3B encoder-decoder UL2 Transformer which produces the desired answer. In such a setup, a contrastively pretrained model provides significantly more useful tokens than one classification pretrained model as in previous PaLI models.

trained to separately embed images and texts, such that a binary classifier using the sigmoid cross-entropy of the dot product of image and text embeddings correctly classifies whether the respective image and text correspond to each other or not. This is similar to CLIP (Radford et al., 2021) and ALIGN (Jia et al., 2021), but was shown to be more efficient, scalable, and robust (Zhai et al., 2023). This is done in order to pretrain the ViT image embedding component, hence the text embedding transformer is discarded when inserting the ViT into PaLI.

**Full PaLI model**   The outputs of the ViT image encoder before pooling form the visual tokens, which are linearly projected and prepended to the embedded input text tokens. Together, these tokens are passed into a pretrained 3B parameter UL2 encoder-decoder language model (Tay et al., 2023), which generates text output. The text input to the model typically consists of a prompt that describes the type of task (e.g., "*Generate the alt_text in* ⟨lang⟩ *at* ⟨pos⟩" for captioning tasks) and encode necessary textual input for the task (e.g., "*Answer in* ⟨lang⟩*: {question}* " for VQA tasks).

### 3.2   STAGES OF TRAINING

The training procedure is similar to that of PaLI and PaLI-X and consists of multiple stages:

**Stage 0: Unimodal pretraining.**   The image encoder is pretrained contrastively on image-text pairs from the web, following the SigLIP training protocol (Zhai et al., 2023). This differs from PaLI and PaLI-X, where a JFT classification pretrained encoder was used. We use a model-based filtering approach similar to that in Schuhmann et al. (2021) that retains about 40% of the pairs. The image encoder is trained at resolution 224×224. The text encoder-decoder is a 3B UL2 model trained following the mixture of denoisers procedure described by Tay et al. (2023).

**Stage 1: Multimodal training.**   Here, the image encoder is combined with the text encoder-decoder as described earlier and in Figure 1. Then, this combined PaLI model is trained on a multimodal task and data mixture, albeit keeping the image encoder `frozen` and using its native (224×224) resolution. The main mixture component is again derived from the WebLI dataset by heuristic filtering of the text quality and using the SplitCap training objective (Chen et al., 2023b). Further ingredients inherited from (Chen et al., 2023b) are multilingual captioning on CC3M-35L and WebLI OCR, cross-lingual VQA and VQG using VQ²A-CC3M-35L, object-aware VQA, as well as object detection. Notably, we do not include task or data derived from video (this was done

Table 1: Performance comparison between contrastively pre-trained ("SigLIP") models and classification pre-trained ("Classif") ViT image encoders using the same PaLI setup, across a wide range of tasks. While linear classification few-shot probing (first column) suggests SigLIP encoders are worse across many tasks, when plugged into PaLI and transferred, they show clear improvements. On the most complicated and detailed image understanding tasks, SigLIP models outperform Classif models by a large margin. Captioning numbers are CIDEr scores, where XM3600 shows the English performance in the first column, and the average across other languages in the second column. RefCOCO numbers are mIoU scores (details in Section 4.3).

| | | **Probe** | **Captioning** | | **VQA** | | | **RefCOCO** | | |
|---|---|---|---|---|---|---|---|---|---|---|
| | | 8 tasks | COCO | XM3600 | v2 | OK | Text | val | + | g |
| G/14 | Classif | 88.1 | 139.9 | 94.5 | 44.7 | 76.7 | 57.2 | 31.9 | 51.6 | 43.5 | 43.4 |
| | SigLIP | **-2.5** | **+0.4** | **+1.6** | **+0.7** | **+0.8** | **+1.4** | **+18.7** | **+15.1** | **+19.1** | **+17.7** |
| L/16 | Classif | 86.2 | 132.6 | 93.0 | 42.3 | 73.7 | 55.6 | 24.9 | 46.9 | 38.8 | 38.8 |
| | SigLIP | **-2.8** | **+3.2** | **+1.4** | **+1.4** | **+1.9** | **+1.9** | **+16.2** | **+17.4** | **+20.9** | **+20.1** |
| B/16 | Classif | 83.7 | 127.7 | 91.7 | 40.7 | 72.3 | 54.7 | 22.5 | 46.3 | 38.1 | 38.4 |
| | SigLIP | **-2.6** | **+3.6** | **-2.0** | **-0.2** | **+1.4** | **+0.9** | **+13.3** | **+16.8** | **+19.6** | **+19.3** |

in PaLI-X), but PaLI-3 retains competitive performance on these benchmarks thanks to its strong image encoder. We do, however, further improve document and text understanding capabilities by enriching WebLI with PDF documents with dense text and web-images described as posters or documents, in over 100 languages.

**Stage 2: Resolution increase.** High-resolution input is a widely accepted way of increasing performance, both due to making more details from the image perceptible, and due to increasing model power via increased sequence length. We increase PaLI-3's resolution by fine-tuning the whole model (unfreezing the image encoder) with a short curriculum of increasing resolutions, keeping checkpoints at 812×812 and 1064×1064 resolution. The data mixture focuses on the part that involves visually-situated text and object detection.

**Task specialization (transfer).** Finally, for each individual task (benchmark), we fine-tune the PaLI-3 model with frozen ViT image encoder on the task's training data as described in the corresponding section. For most tasks, we fine-tune the 812×812 resolution checkpoint, but for two document understanding tasks, we go up to 1064×1064 resolution.

## 4 EXPERIMENTS

### 4.1 CLASSIFICATION OR CONTRASTIVELY PRETRAINED VIT?

We first perform a controlled comparison of different ViT models within the PaLI framework. We consider two types of ViT models: classification pretrained ("Classif") on the JFT dataset and contrastively pretrained on the WebLI dataset ("SigLIP"). We perform these experiments using a fixed 224×224 resolution (i.e. only include Stage 1) to save compute. We further shorten the Stage 1 phase to 20% of the full PaLI-3 schedule used in the remainder of this paper.

The results in Table 1 paint a clear picture overall: While the few-shot linear classification (Dosovitskiy et al., 2021) of SigLIP models falls behind, when used in PaLI-3, SigLIP models provide moderate gains on "simpler" tasks such as captioning and question-answering, and large gains for more "complicated" scene-text and spatial understanding tasks such as TextVQA and RefCOCO variants. This motivates the departure from classification pretrained image encoders, and switching to sigmoid-contrastively pretrained ones for building PaLI-3.

Table 2: Results on benchmarks more focused on understanding visually-situated text. TextCaps, TextVQA, STVQA, InfographicVQA and DocVQA are all evaluated using the corresponding official evaluation server. Methods marked by "†" are trained on additional VQA data similar to the target benchmark, before finetuning on the target benchmark. For ChartQA, we compare with similar setups by finetuning without chain-of-thought or similar prompting techniques. The SOTA models are (a) Chen et al. (2023a), (b) Powalski et al. (2021), (c) Peng et al. (2022).

| Model | Text Caps | Text VQA | ST VQA | OCR VQA | Info VQA | Doc VQA | AI2D | Chart QA | Screen2 Words | Widget Cap | Avg. of first 8 |
|---|---|---|---|---|---|---|---|---|---|---|---|
| *with OCR pipeline input* | | | | | | | | | | | |
| SOTA | 163.7 (a) | **80.78** (a) | 84.5 (a) | 77.3 (a) | 61.2 (b)† | 88.4 (c)† | **81.4** (a) | **72.3** (a) | - | - | 88.7 |
| PaLI-3 | **164.3** | **80.78** | **85.7** | **77.8** | **62.4** | **88.6** | 75.2 | 69.5 | - | - | 88.0 (-0.7) |
| *without OCR pipeline input* | | | | | | | | | | | |
| SOTA | 147.0 (a) | 71.44 (a) | 79.9 (a) | 75.0 (a) | 49.2 (a) | 80.0 (a) | **81.2** (a) | **70.9** (a) | 127.9 (a) | 153.0 (a) | 81.8 |
| PaLI-3 | **158.8** | **79.51** | **84.1** | **76.7** | **57.8** | **87.6** | 75.2 | 70.0 | **130.7** | **159.8** | 86.2 (+4.4) |

## 4.2 VISUALLY-SITUATED TEXT UNDERSTANDING

We evaluate PaLI-3 on visually-situated text understanding tasks: TextCaps (Sidorov et al., 2020), TextVQA (Singh et al., 2019), STVQA (Biten et al., 2019), OCRVQA (Mishra et al., 2019), InfographicVQA (Mathew et al., 2022), DocVQA (Mathew et al., 2021), ChartQA (Masry et al., 2022), Scree2Words (Wang et al., 2021a), and WidgetCap (Li et al., 2020). The images in those datasets span a wide range of domains such as natural images, illustrations, documents and user interfaces.

For the InfographicVQA and DocVQA benchmarks we fine-tune the 1064×1064 resolution model, all others use the 812×812 one. We report the standard metrics for each benchmark, namely: CIDEr score for all the Image Captioning benchmarks; VQA accuracy for VQAv2, OKVQA, and TextVQA; Average Normalized Levenshtein Similarity (ANLS) for ST-VQA, DocVQA and InfographicsVQA; Exact match (EM) for TallyQA, OCR-VQA and AI2D; Relaxed-Accuracy (RA) for ChartQA. For visually-situated text understanding, external OCR systems are usually leveraged to provide OCR annotations of the image as and additional input to the model for boosting performance. Here we follow (Chen et al., 2023a) and report the result of finetuning PaLI-3 both with and without OCR inputs. The OCR annotations are obtained using the same service as that for the training set. As shown in Table 2, PaLI-3 achieves SOTA performance on a vast majority of the captioning and VQA benchmarks both with and without external OCR input. The exception is AI2D and ChartQA, which require not just understanding but also strong reasoning capability over diagrams and charts, respectively. For these two benchmarks, PaLI-3 falls slightly behind PaLI-X (Chen et al., 2023a) likely due to the latter's significantly larger 32B LLM being better at reasoning.

Averaging over the 8 benchmarks in Table 2 that have results in all settings, PaLI-3 is only 0.7 points behind all SOTA methods combined in the setting where external OCR systems are used. However, in the setting without such external system, PaLI-3 has a significant 4.4 point advantage over all SOTA methods combined. For TextCaps, TextVQA, InfographicVQA and DocVQA this advantage is 8 points or more. Finally, we can see that PaLI-3 without any external OCR system is only 1.8 points behind relying on one, suggesting the image encoder learns a strong intrinsic OCR capability.

## 4.3 REFERRING EXPRESSION SEGMENTATION

We extend PaLI-3 with the capability to predict segmentation masks via language-like output. To this end, we make use of the vector-quantized variational auto-encoder (VQ-VAE) from Ning et al. (2023). The VQ-VAE is trained to learn a discrete codebook of 128 mask tokens. Its encoder can tokenize a 64 × 64 pixels segmentation mask into 16 mask tokens, which its decoder can convert back. We train PaLI-3 to predict a single segmentation mask. First, PaLI-3 outputs 4 coordinates as text, representing a bounding box. This is followed by 16 mask tokens that represent the mask inside the bounding box.

Table 3: PaLI referring expression segmentation results on RefCOCO (Yu et al., 2016) variants. All results are mIoU on the `val` split.

| Model | RefCOCO | RefCOCO+ | G-Ref |
|---|---|---|---|
| RefTr (Li & Sigal, 2021) | 74.34 | 66.75 | 66.63 |
| PolyFormer (Liu et al., 2023) | 76.94 | 72.15 | 71.15 |
| PaLI-3 (Ours) | **77.33** | **73.53** | **72.72** |

We fine-tune PaLI-3 on the combined training sets of RefCOCO, RefCOCO+, and RefCOCOg (Yu et al., 2016)[2] at 812×812 resolution. Each training example consists of a referring expression (e.g. "the large brown dog on the left"), and a box with segmentation mask. We prompt PaLI with the prompt "*detect: the large brown dog on the left* ⟨extra_id_0⟩", and the target is a sequence like "*348 543 684 664* ⟨mask_token_81⟩ . . . ⟨mask_token_10⟩". The target sequence contains 16 mask tokens between 0 and 127 that are generated by the VQ-VAE encoder using the segmentation mask cropped and resized to $64 \times 64$ as input.

The results in Table 1 demonstrate that contrastive pretraining is much more effective than classification pretraining for localization task of this type. Table 3 shows that the full PaLI-3 model is able to slightly outperform the state of the art in referring expression segmentation.

## 4.4 NATURAL IMAGE UNDERSTANDING

In this section, we evaluate PaLI-3 on general vision-language understanding tasks, including COCO captions (Karpathy & Fei-Fei, 2015) and VQAv2 (Goyal et al., 2017) which target general visual understanding, OKVQA (Marino et al., 2019) which focuses on knowledge-based understanding and TallyQA (Acharya et al., 2019) which measures performance on counting under textual guidance. All results for the benchmarks presented in Table 4 use 812×812 resolution. As in previous work, they employ no external OCR module, since these benchmarks rarely involve text in the image.

Overall, PaLI-3 shows very strong performance on these benchmarks despite its significantly smaller size compared to recent SOTA models. For COCO, PaLI-3 outperforms all models but BEiT-3 and the 17B and 55B PaLI. On VQAv2 and TallyQA, PaLI-3 exceeds all previous models except PaLI-X, with a less than 1 point gap on VQAv2. For the knowledge-intensive OKVQA task, which usually benefits from a large language component, PaLI-3 is only behind PaLM-E (562B) and PaLI-X (55B) but still outperforms the 32-shot Flamingo (80B) model.

Table 4: Results on COCO Captions (Karpathy split), VQAv2, OKVQA, and TallyQA. (*Flamingo reports 32 shot result). Underscored numbers indicate that PaLI-3 is only behind the 55B PaLI-X and is better than all other models in the list.

| Model | COCO Karp.-test | VQAv2 test-dev | VQAv2 test-std | OKVQA val | TallyQA Simple | TallyQA Complex |
|---|---|---|---|---|---|---|
| SimVLM | 143.3 | 80.03 | 80.34 | - | - | - |
| CoCa (2.1B) | 143.6 | 82.3 | 82.3 | - | - | - |
| GIT (0.7B) | 144.8 | 78.56 | 78.81 | - | - | - |
| GIT2 (5.1B) | 145.0 | 81.74 | 81.92 | - | - | - |
| OFA (0.9B) | 145.3 | 82.0 | 82.0 | - | - | - |
| Flamingo (80B) | 138.1 | 82.0 | 82.1 | 57.8* | - | - |
| BEiT-3 (1.9B) | 147.6 | 84.2 | 84.0 | - | - | - |
| PaLM-E (562B) | 138.7 | 80.0 | - | **66.1** | - | - |
| MoVie | - | 69.26 | - | - | 74.9 | 56.8 |
| PaLI-17B | 149.1 | 84.3 | 84.3 | 64.5 | 81.7 | 70.9 |
| PaLI-X (55B) | **149.2** | **86.0** | **86.1** | **66.1** | **86.0** | **75.6** |
| PaLI-3 (5B) | 145.9 | 85.0 | 85.2 | 60.1 | 83.3 | 70.5 |

---

[2]We removed all validation and test images from the training set for both PaLI and the VQ-VAE

Table 5: Results for Video Captioning and Video-QA using up to 16 frames. †GIT2 directly optimizes the CIDEr metric. mPLUG-2 is Xu et al. (2023), PaLI-X is Chen et al. (2023a), GIT2 is Wang et al. (2022a), and Flamingo-32 is the 32-shot variant of Alayrac et al. (2022).

| | MSR-VTT | | Activity-Net | | VATEX | SMIT | NExT-QA |
|---|---|---|---|---|---|---|---|
| Method | Caption | QA | Caption | QA | Caption | Caption | QA |
| Prior SOTA | **80.3** | 48.0 | **54.9** | 49.4 | 94.0† | **43.5** | **38.3** |
| | mPLUG-2 | mPLUG-2 | PaLI-X | PaLI-X | GIT2 | PaLI-X | Flamingo-32 |
| PaLI-3 | 78.3 | **49.3** | 50.8 | **51.2** | 66.9 | 39.6 | 37.7 |

## 4.5 Video Captioning and Question Answering

We fine-tune and evaluate the PaLI-3 model on 4 video captioning benchmarks: MSR-VTT (Xu et al., 2016), VATEX (Wang et al., 2019), ActivityNet Captions (Krishna et al., 2017), and Spoken Moments in Time (Monfort et al., 2021). We do the same for 3 video question-answering benchmarks: NExT-QA (Xiao et al., 2021), MSR-VTT-QA (Xu et al., 2017), and ActivityNet-QA (Yu et al., 2019). A brief description of each benchmark and its usage is provided in Appendix A.

Following the setup from PaLI-X (Chen et al., 2023a), we fine-tune our model using the Stage 1 checkpoint with 224×224 resolution for each task separately. We sample at most 16 frames with a fixed temporal stride for each benchmark. Each frame is independently processed by the ViT image encoder, the resulting visual tokens are simply concatenated, leading to up to 4096 visual tokens. A key difference from the PaLI-X setup is that there is no video data in PaLI-3 pretraining, meaning PaLI-3 has never seen multi-frame inputs during pretraining.

Despite not being pretrained with video data, PaLI-3 achieves excellent video QA results with a small model size: a new state of the art performance on MSR-VTT-QA and ActivityNet-QA, and competitive results on NextQA. The consistent improvements on image and video QA highlight the benefits of adopting the contrastive ViTs. PaLI-3 also achieves respectable video captioning results, under-performing the SOTA by only 3 CIDEr points on average. Considering the model size, PaLI-3 appears to be an excellent choice in terms of both performance and practicality.

## 4.6 Direct image encoder evaluation

Here, we aim to directly evaluate the learned image encoder (ViT-G model) without the surrounding language model, i.e. not the full PaLI-3. All results are summarized in Table 6.

First, we test image classification capabilities using the standard ImageNet (Russakovsky et al., 2014) benchmark and two of its most popular variants (Beyer et al., 2020; Recht et al., 2019). We fine-tune the unimodal image encoder from Stage 0 on ImageNet and compare it to fine-tuning of classification pretrained ViTs used in previous PaLI models. The main comparison is to the classification (Classif) pretrained ViT-G/14 model from Zhai et al. (2022a). The SigLIP slightly

Table 6: Evaluations of the visual component in isolation (without the language model). We report fine-tuned classification accuracy on ImageNet, ImageNet-ReaL and ImageNet-V2; average zero-shot cross-modal retrieval recall@1 across 36 languages on XM3600; average 10-shot linear probe classification accuracy across 8 tasks.

| Model | Encoder | ImageNet (fine-tuning) | | | | XM3600 (retrieval) | | Probe |
|---|---|---|---|---|---|---|---|---|
| | | Res. | Val | ReaL | v2 | I→T | T→I | 8 tasks |
| PaLI-3 | SigLIP ViT-G | 518px | 89.6 | 90.9 | 82.3 | 56.9 | 44.0 | 85.6 |
| PaLI-15B | Classif ViT-G | 518px | 90.5 | 90.8 | 83.3 | - | - | 88.1 |
| PaLI-17B | Classif ViT-e | 644px | 90.9 | 91.1 | 84.3 | 36.0 | 28.5 | 89.5 |
| PaLI-X | Classif ViT-22B | 756px | 89.2 | 91.0 | 83.7 | - | - | 89.9 |

Table 7: RAI statistics for captions generated by PaLI-3 on FairFace (Karkkainen & Joo, 2021).

| | Perceived Gender | | Ethnicity | | | Age Bucket | | | |
| | Lowest | Highest | Lowest | Median | Highest | Lowest | Median | Highest | **Overall** |
|---|---|---|---|---|---|---|---|---|---|
| **Toxicity** | 0.02% | 0.05% | 0.00% | 0.00% | 0.10% | 0.00% | 0.00% | 0.07% | **0.04%** |
| **Profanity** | 0.00% | 0.00% | 0.00% | 0.00% | 0.00% | 0.00% | 0.00% | 0.00% | **0.00%** |
| **Insult** | 0.04% | 0.10% | 0.00% | 0.07% | 0.14% | 0.00% | 0.06% | 0.17% | **0.07%** |
| **Threat** | 0.10% | 0.12% | 0.00% | 0.14% | 0.20% | 0.00% | 0.00% | 0.21% | **0.11%** |
| **Attack** | 0.00% | 0.00% | 0.00% | 0.00% | 0.00% | 0.00% | 0.00% | 0.00% | **0.00%** |

Table 8: RAI score statistics in the captions generated by PaLI-3 on MIAP (Schumann et al., 2021).

| | Perceived Gender | | Age Bucket | | | Skin Tone | | | |
| | Lowest | Highest | Lowest | Median | Highest | Lowest | Median | Highest | **Overall** |
|---|---|---|---|---|---|---|---|---|---|
| **Toxicity** | 0.05% | 0.10% | 0.05% | 0.24% | 0.48% | 0.00% | 0.00% | 0.26% | **0.07%** |
| **Profanity** | 0.10% | 0.10% | 0.00% | 0.12% | 0.24% | 0.00% | 0.00% | 0.10% | **0.10%** |
| **Insult** | 0.10% | 0.14% | 0.00% | 0.13% | 0.17% | 0.00% | 0.00% | 0.38% | **0.13%** |
| **Threat** | 0.34% | 0.80% | 0.30% | 0.54% | 1.68% | 0.00% | 0.59% | 0.94% | **0.65%** |
| **Identity Attack** | 0.00% | 0.00% | 0.00% | 0.00% | 0.00% | 0.00% | 0.00% | 0.00% | **0.00%** |

lags behind in terms of top-1 and v2 accuracy, but matches in terms of ReaL accuracy (Beyer et al., 2020), a metric which avoids measuring "overfitting" to ImageNet peculiarities.

Second, we report multilingual image-text retrieval results on the Crossmodal-3600 benchmark (Thapliyal et al., 2022). Since classification pretrained image encoders do not have this capability, we LiT-tune (Zhai et al., 2022b) a text encoder for it on the multilingual WebLI dataset. We also LiT-tune a new text encoder for the SigLIP image encoder in the exact same setting, to remove any confounders from the comparison. The SigLIP ViT-G model clearly outperforms the classification pretrained larger ViT-e model.

Finally, we perform linear probing of the representation in the few-shot setting following Dosovitskiy et al. (2021); Zhai et al. (2022a) across the 8 different classification tasks also used in Zhai et al. (2022a) and report the average performance. Here, we see that SigLIP lags behind, likely because the representation is not pretrained in a way that supports linear separability, as was recently uncovered by Tschannen et al. (2023).

Taken together, these results paint a clear picture: the best and largest classification pretrained image encoders appear (slightly) better when evaluated on standard classification tasks, however they are significantly worse than SigLIP pretrained image encoders for vision-language tasks.

## 5 MODEL FAIRNESS, BIASES, AND OTHER POTENTIAL ISSUES

We follow the evaluation protocol of Chen et al. (2023a) to assess the model fairness, biases, and other potential issues. First, we use the MIAP (Schumann et al., 2021) and FairFace (Karkkainen & Joo, 2021) datasets to generate captions and use the Perspective API (Lees et al., 2022) (threshold $> 0.8$) to measure toxicity and profanity among other potential issues. Table 7 (FairFace) and Table 8 (MIAP) summarize the results. Slices with $< 20$ examples are dropped from the analysis. Overall, we observe a low level of toxicity and profanity among others, across all slices. The results are comparable to those in PaLI-X (Chen et al., 2023a).

Second, we examine the level of demographic parity (Dwork et al., 2012) in the model itself. Following (Chen et al., 2023a), we feed an image from the CelebA dataset (Liu et al., 2015) into PaLI-3 with the chosen occupation as a prefix and record the average log-perplexity score of the model generation. The demographic parity is the difference between average log-perplexity within the demographic groups. Figure 2 summarizes the results: Similar to PaLI-X, PaLI-3 tends to assign a *higher* log-perplexity score to women than men across most occupations with a mean difference of $\mu = 0.37$. However, fewer occupations in PaLI-3 fall outside the interval $\mu \pm 2\sigma$ compared to PaLI-X.

Table 9: Detection error rate for "person" in PaLI-3 using the subset of the MIAP dataset (Schumann et al., 2021) that contain exactly a single individual in the image. PaLI-3 maintains a low error rate across all subgroups. Skin tone follows the Monk Skin Tone Scale (Monk, 2019). Numbers inside square brackets correspond to the size of each bucket.

| Skin Tone | 1 [2] | 2 [871] | 3 [3008] | 4 [522] | 5 [184] | 6 [85] | 7 [54] | 8 [49] | 9 [6] | 10 [1] |
|---|---|---|---|---|---|---|---|---|---|---|
| | 0.00% | 0.00% | 0.17% | 0.39% | 0.00% | 0.00% | 0.00% | 0.00% | 0.00% | 0.00% |
| **Gender** | **Predominantly Feminine** [2437] | | | | **Predominantly Masculine** [3544] | | | | | |
| | 0.41% | | | | 1.78% | | | | | |
| **Age Bucket** | **0-2 yrs** [17] | | **3-19 yrs** [568] | | **20-59 yrs** [4925] | | **> 60 yrs** [247] | | | |
| | 0.00% | | 0.18% | | 1.30% | | 0.82% | | | |

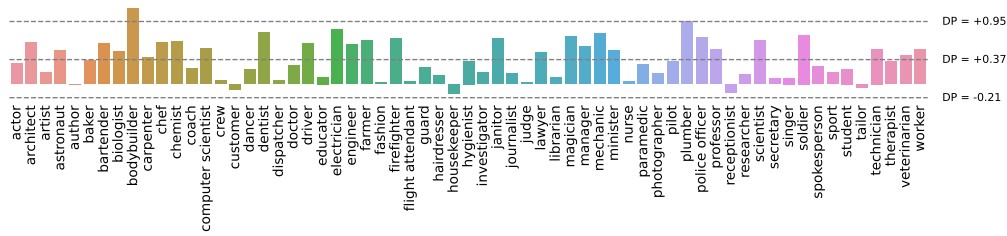

Figure 2: Level of demographic parity (DP) in PaLI-3's output for CelebA images, comparing the average log-perplexity between females and males. Values close to zero indicate absence of bias. The dotted lines correspond to $\mu$ and $\mu \pm 2\sigma$. DP for all occupations falls within the 95% confidence interval, except "bodybuilder" and "plumber" which the model tends to strongly associate with men.

Third, we compare performance across all subgroups on a detection task using the MIAP dataset, following again (Chen et al., 2023a). For images containing exactly a single individual, we query PaLI-3 with the question: "Is there a person in this image?" and evaluate the accuracy of its response. Table 9 summarizes the results. The error rate (false negatives) is very low across all subgroups.

For analysis of the WebLI dataset itself, such as the correlations between perceived gender and occupations, see Chen et al. (2023a).

**Limitations.** The limitations of this work are very similar to those already presented in the literature. We refer to (Chen et al., 2023a), which raises all limitations that apply to our work.

## 6 CONCLUSION

In this paper, we took a closer look at the pretraining of the image encoder in large VLMs, specifically the PaLI type of model. By performing controlled experiments on that part, for the first time, we clearly compare the two camps of classification pretraining and image-text (contrastive) pretraining, finding that the latter can lead to better and more efficient VLMs, especially for localization and text understanding tasks. This is just *one small aspect* of VLMs, and we hope this study and result spurs to further detailed investigations of the *many other aspects* of training VLMs.

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
