# A    ADDITIONAL RESULTS: VIDEO CAPTIONING AND QA

## A.1    DATASETS & BENCHMARKS

Because not all videos in the benchmarks are available online at the time of experimentation when freshly collected the data, the effective numbers of videos is smaller than the public official splits in some cases. Table 10 reports the details numbers for the subset used in our training and evaluation. We follows the same experimental settings used in PaLI-X (Chen et al., 2023a), including the dataset splits configuration and the evaluation metrics. Please refer to PaLI-X (Chen et al., 2023a) for more data-related details.

|  |  | MSR-VTT | VATEX | ANet-Cap | SMIT | M-V-QA | . ANet-QA | NExT-QA |
|---|---|---|---|---|---|---|---|---|
| Original size | valid. | 497 | 3000 | 17505 | 14604 | 12278 | 18000 | 5343 |
|  | test | 2990 | 6000 | 17031 | 3513 | 72821 | 8000 | 9178 |
| Dataset size | valid. | 325 | 2646 | 14566 | 8096 | 8160 | 10000 | 5343 |
|  | test | 2135 | 5242 | 14197 | 3513 | 52623 | 7040 | 9178 |
| % Remaining | valid. | 65.39 | 88.20 | 83.21 | 100.00 | 66.46 | - | 100.00 |
|  | test | 71.40 | 87.37 | 83.36 | 100.00 | 72.26 | 88.00 | 100.00 |

Table 10: As we freshly collect the data sets, the actual amount of training data is smaller than the public benchmarks, making the tasks more challenging. Except for NextQA and SMIT, there are more than 10% of the videos missing in both training and evaluation.

# B    ADDITIONAL RESULTS: CROSSMODAL-3600 RETRIEVAL

We present more results of zero-shot image-text retrieval results (recall@1) on Crossmodal-3600 (Thapliyal et al., 2022) in this section. Detailed results across 36 languages for SigLIP ViT-G and Classif ViT-e are presented in Figure 3, Figure 4 and Table 11.

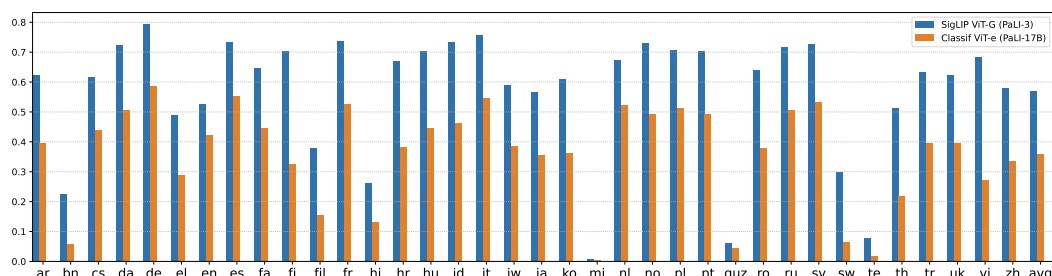

Figure 3: Image-to-text zero-shot retrieval recall@1 on crossmodal-3600 for SigLIP ViT-G and Classif ViT-e.

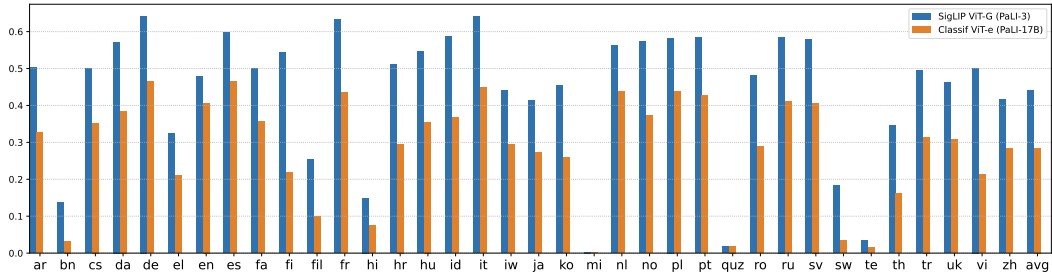

Figure 4: Text-to-image zero-shot retrieval recall@1 on crossmodal-3600 for SigLIP ViT-G and Classif ViT-e.

| Language | Image-to-text | | Text-to-image | |
|---|---|---|---|---|
| | SigLIP ViT-G | Classif ViT-e | SigLIP ViT-G | Classif ViT-e |
| ar | 62.22 | 39.69 | 50.33 | 32.60 |
| bn | 22.67 | 5.67 | 13.61 | 3.31 |
| cs | 61.69 | 44.03 | 50.10 | 35.24 |
| da | 72.47 | 50.75 | 57.16 | 38.48 |
| de | 79.28 | 58.53 | 64.20 | 46.50 |
| el | 49.11 | 29.03 | 32.36 | 20.92 |
| en | 52.64 | 42.11 | 47.89 | 40.63 |
| es | 73.31 | 55.22 | 59.77 | 46.55 |
| fa | 64.56 | 44.50 | 50.09 | 35.58 |
| fi | 70.39 | 32.64 | 54.34 | 21.80 |
| fil | 37.89 | 15.53 | 25.40 | 10.04 |
| fr | 73.81 | 52.61 | 63.28 | 43.47 |
| hi | 26.22 | 13.14 | 14.94 | 7.42 |
| hr | 66.89 | 38.31 | 51.03 | 29.55 |
| hu | 70.22 | 44.67 | 54.63 | 35.49 |
| id | 73.53 | 46.33 | 58.62 | 36.75 |
| it | 75.56 | 54.53 | 64.14 | 44.76 |
| iw | 59.14 | 38.67 | 44.10 | 29.39 |
| ja | 56.69 | 35.47 | 41.31 | 27.24 |
| ko | 61.03 | 36.11 | 45.50 | 25.95 |
| mi | 0.64 | 0.33 | 0.30 | 0.22 |
| nl | 67.25 | 52.14 | 56.23 | 43.79 |
| no | 73.03 | 49.17 | 57.40 | 37.35 |
| pl | 70.69 | 51.42 | 58.24 | 43.72 |
| pt | 70.22 | 49.19 | 58.47 | 42.73 |
| quz | 6.28 | 4.31 | 1.89 | 1.90 |
| ro | 63.92 | 37.75 | 48.20 | 28.82 |
| ru | 71.69 | 50.64 | 58.51 | 41.11 |
| sv | 72.69 | 53.22 | 57.76 | 40.66 |
| sw | 29.86 | 6.42 | 18.31 | 3.41 |
| te | 7.81 | 1.92 | 3.39 | 1.42 |
| th | 51.14 | 22.00 | 34.65 | 16.06 |
| tr | 63.36 | 39.50 | 49.41 | 31.47 |
| uk | 62.19 | 39.53 | 46.15 | 30.81 |
| vi | 68.50 | 27.08 | 50.14 | 21.28 |
| zh | 57.92 | 33.61 | 41.51 | 28.24 |
| avg | 56.85 | 35.99 | 43.98 | 28.46 |

Table 11: Crossmodal-3600 zero-shot retrieval recall@1 (%) for SigLIP ViT-G (2B params) and Classif ViT-e (4B params). SigLIP ViT-G is significantly better than Classif ViT-e across all the languages. SigLIP improves from 28.5% to 44.0% on average for the text-to-image retrieval task.