# OpenReview forum: "PaLI-3 Vision Language Models: Smaller, Faster, Stronger"
_ICLR.cc/2024/Conference — Submitted to ICLR 2024_

### Official Review · Reviewer_LxB9 · 2023-10-13

**Soundness:** 4 excellent
**Presentation:** 4 excellent
**Contribution:** 4 excellent
**Rating:** 8
**Confidence:** 4

**Summary:**

This paper introduces PaLI-3, a new vision language model. One important modification is using SigLIP as the vision module encoder. The final model is small but effective. It performs comparably with many larger models on various benchmarks and also achieves better results in localization and text understanding than prior works.

**Strengths:**

1) Very good results in terms of cost-effectiveness trade-off.Comprehensive evaluation on various benchmarks.
2) The paper is very easy to read and understand.
3) The approach is simple and easy to implement.
4) The effectiveness of SigLIP is very insightful. It seems that such a simple modification can give a significant improvement. It shows the potential of the importance of designing a smarter training objective that aligns better with the language models.

**Weaknesses:**

1) I strongly encourage authors to provide more comparisons between CLIP and SigLIP under this paper's setting. The current ablation only includes the comparison between SigLIP and vanilla classification.
2) I understand the paper mainly focuses on a smaller and cheaper model, as stated in the title. However, I think it is important to study the scaling results to check the effectiveness on a larger scale. Can the SigLIP still be so effective when using a larger vision encoder and language models? It would also be interesting to further scale down the model and see what would happen.

**Questions:**

1) Any plan to release the code? UL2 and SigLIP are both open-sourced. I think it would be nice to have an open-sourced version (would be better to use open-sourced data to pretrain) and will be easy to compare and use this model as a baseline. This model is small. If you can provide a fully reproducable version, I believe more folks from GPT-poor institutes would be motivated to follow.
2) I understand this paper's setting is not a huge model with a very strong zero-shot ability like GPT-4V. However, I'm highly interested in what do authors think about the potential of using SigLIP as the vision encoder for real LLMs (maybe > 100B or so).
3) Could you provide a detailed discussion about the difference between PaLI-3 and the previous versions? It would be better to have a table and show the differences directly.

---

> ### Author Response · Authors · 2023-11-22
> **Response to Official Review of Submission7032 by Reviewer LxB9**
>
> **[W1]**
>
> We thank the reviewer for the suggestion. We would like to note that we do not claim contrastively pretrained visual encoder is always superior to a classification-pretrained one for all purposes. Instead, we reported our observations on contrastively trained SigLIP that “despite SigLIP slightly underperforming on image classification, it performs much better on Visual Language tasks.” (See our contribution #1) with a focus on the relative advantage of contrastively trained models on VL tasks, which we believe holds true for similarly trained models including OpenCLIP. That said, we agree with the reviewer that the comparison to OpenCLIP may provide a complementary datapoint for the community. We are working on setting up and performing such a comparison and are trying to add an additional baseline based on OpenCLIP before the final version of the draft.
>
>
> **[W2]**
>
> We agree with the reviewer that it is important to study the scaling laws to check the effectiveness on a larger scale. The result in Table 1 is about the scaling behavior, where we first tested SigLIP sizes B/16 and L/16, and the promising scaling behavior was exactly what drove us to push SigLIP to 2B (G/14) parameter scale used for PaLI-3.
>
> **[W3]**
>
> We thank the reviewer for this suggestion. Please see the two main points below. We will expand and add to the revised paper.
> - PaLI-3 used SigLIP pretrained ViT, while previous versions used the classification pretrained ones.
> - The training mixture of PaLI-3 is based on the learnings of PaLI-X, with a few key new components, including the further filtered and refined WebLI dataset, especially on the OCR data, and we curated and added web-scale document images in the training.
>
> **[Q1] Code release**
>
> We are actively working on a potential open-sourcing of our model, however there are still aspects related to the open-sourcing effort that we are still figuring out.
>
> **[Q2] Using SigLIP for much larger LLMs**
>
> We thank the reviewer for the suggestion. Indeed, we are also very interested in the potential of SigLIP in large LLMs. We plan to explore this in the followup work with the potentially further scaled-up SigLIP variants.
>
> **[Q3] Difference between PaLI-3 and previous versions. Similar to W3**
>
> We thank the reviewer for this suggestion. Please see what we listed above, and we will expand and add to the revised paper

---

### Official Review · Reviewer_Cwas · 2023-10-31

**Soundness:** 3 good
**Presentation:** 3 good
**Contribution:** 2 fair
**Rating:** 3
**Confidence:** 5

**Summary:**

This paper presents the latest improvement in the so-called PALI series of VL models. The main contributions are the replacement of the Visual Encoder with a SigLIP model and the increase of resolution. The authors present many experimental results which show the effectiveness of their pipeline.

** Post rebuttal: ** The fact that contrastively pre-trained ViT with language supervision on billion scale dataset outperforms purely visually trained encoders is a well known fact in literature. All methods proposed in last couple of years (e.g. Flamingo, BLIP2) use that. Moreover, just replacing a backbone with another backbone is not of sufficient novelty for ICLR. Finally, increasing accuracy by increasing resolution is also well known. So unfortunately I will sit on my original rating.

**Strengths:**

The main strength of the paper is the numerous experiments the authors have carried out and the good results presented. Moreover, the paper is fairly easy to follow.

**Weaknesses:**

Unfortunately I don't believe that the claimed contributions (used of SigLIP and increase of resolution) are enough for ICLR. The finding that contrastively pre-trained visual backbone with language supervision works better than training for classification doesn't seem very surprising. Moreover, training follows previous PALI training pipelines so no particular novelty in this regard either. Actually incorporating these improvements could probably benefit any other model compared with the proposed one.

**Questions:**

No questions

---

> ### Author Response · Authors · 2023-11-22
> **Response to Official Review of Submission7032 by Reviewer Cwas**
>
> **[Regarding the first point from the reviewer]**
>
> It might sound intuitive to the reviewer, but to date, there was no direct comparison published, and by looking at individual papers in isolation, it may have seemed that JFT-pretraining leads to superior combined models, given PaLI (arXiv:2209.06794) and PaLI-X (arXiv:2305.18565) results.
>
> In this work we actually perform apples-to-apples comparison using contrastively trained SigLIP ViTs and discriminatively trained ViTs with the same architecture and and quantify the wins and losses. Our comprehensive experiments show that **there is not a simple winner**: while the contrastively pretrained SigLIP models still lose to the discriminatively trained counterparts on visual classification tasks, SigLIP-based PaLI-3 shows superior performance on Vision-Language tasks. We believe that our experiments provide concrete data points and insights to the community beyond just intuitions.
>
> **[Regarding the second point from the reviewer]**
>
> In this work, we made substantial improvements over the PaLI training recipe. Our contributions to the training mixture is generally applicable. Even in the reviewer’s own hypothetical statements, “Actually incorporating these improvements could probably benefit any other model compared with the proposed one.”, the reviewer acknowledged the significance of our training recipe.
>
> It was very surprising to us as authors that such a compact model (at only 5B) was able to outperform a related but much larger-capacity model (PaLI-X at 55B) across so many tasks. One of the goals of this paper was to highlight what parts would be responsible for such a surprising result, and we consider that the research community would be equally interested in understanding it.

---

### Official Review · Reviewer_fT87 · 2023-11-02

**Soundness:** 3 good
**Presentation:** 3 good
**Contribution:** 2 fair
**Rating:** 6
**Confidence:** 4

**Summary:**

This paper presents PaLI-3, a vision-language model with only 5B parameters but achieves state-of-the-art results across several benchmarks. The authors begin by comparing contrastively pretrained visual encoders and classification-pretrained ViT models, drawing the conclusion that the contrastively pretrained visual encoder demonstrates better performance on vision-language tasks, especially grounding tasks. Compared to previous state-of-the-art models (SOTAs), PaLI-3 can achieve competitive scores with significantly fewer overall parameters. Despite the training process being conducted without any video inputs, PaLI-3 is still capable of accomplishing video-based tasks.

**Strengths:**

- The conclusion that a contrastively pretrained visual encoder can outperform a classification-pretrained encoder in vision-language tasks, particularly in grounding, is valuable and beneficial to the vision-language community.
- Strong performance with much less parameters.
- Sufficient in-depth analysis on general tasks and fairness, bias and potential issues are performed to better model understanding.

**Weaknesses:**

- The main weakness of PaLI-3, from my perspective, is the way the authors used to draw their conclusion. Specifically, the authors claim that because SigLIP shows better performance than the classification-pretrained visual encoder used by PaLI and PaLI-X, they conclude that a contrastively pretrained visual encoder is superior to a classification-pretrained one. However, it's worth noting that most of the accessible contrastively pretrained visual encoders for the vision and vision-language community are members of the OpenCLIP family. Have you ever attempted to utilize OpenCLIP as a vision encoder?
- The results in Section 4 are per-benchmark finetuned. What's the performance of PaLI-3 without task-specific fine-tuning (zero-shot)? Is it possible to generate target answers with few-shot demonstrations by prompting (in-context learing)?
- As mentioned in Section 3.2, during stage 0 of PaLI-3's training process, the contrastive visual encoder is pretrained with a 3B UL2 as a text encoder-decoder. Subsequently, the same 3B UL2 model is employed as the language model for PaLI-3. Is this consistency in using the same language model for both contrastive visual pre-training and generative vision-language pre-training crucial or not? Have any experiments been conducted on this?

**Questions:**

See weaknesses

**Details Of Ethics Concerns:**

N/A.

---

> ### Author Response · Authors · 2023-11-22
> **Response to Official Review of Submission7032 by Reviewer fT87**
>
> **[W1]**
>
> We thank the reviewer for the suggestion. We would like to note that we do not claim that “contrastively pretrained visual encoder is superior to a classification-pretrained one for all purposes”. Instead, we reported our observations on contrastively trained SigLIP that “despite SigLIP slightly underperforming on image classification, it performs much better on Visual Language tasks.” (See our contribution #1) with a focus on the relative advantage of contrastively trained models on VL tasks, which we believe holds true for similarly trained models including OpenCLIP.
>
> For OpenCLIP, We have not tried it at the time of submission, since SigLIP is also open-source and significantly better across benchmarks. However, we agree this can be an interesting ablation, for which we have started to set up and are trying to add an additional baseline based on OpenCLIP before the final version of the draft.
>
> **[W2]**
>
> We expect the zero-shot performance to be relatively poor, since there is no task aimed for this in the pre-training mixture, and hence the model cannot know "the coco vocabulary" from which it should generate. PaLI-3 is also not a chatbot, but rather a (compact) foundation model capable to be transferred to various tasks.
>
> Meanwhile, few-shot in-context ability is a good idea that we could and likely should add in the pre-training mixture, and would provide an in-between alternative between fine-tuning and zero-shot.
>
> **[W3]**
>
> Thanks for the question, it highlights an unclarity in the write-up. The stage0 pre-training does not use a UL2 encoder-decoder. It is trained with a transformer encoder for the text with a similar size to ViT-L, following SigLIP (and like CLIP), which we then discard in the following stages. Thus, no such consistency in the language model is required. We will emphasize this aspect better in the paper’s next version.

---

### Meta-Review · Area_Chair_vFCQ · 2023-12-11

**Metareview:**

This paper introduces PaLI-3, a new model in the PaLI series, which sets new state-of-the-art (SOTA) benchmarks in several vision-language tasks with a relatively small parameter count. The authors present extensive ablation studies, highlighting a refined training approach and design for vision-language (VL) models. A key finding is the superiority of contrastively trained visual encoders over those pretrained for classification in VL tasks.


Reviewer LxB9 recognizes PaLI-3's empirical strength and simplicity, while suggesting additional comparative analysis with CLIP and SigLIP to bolster claims about the superioiry of contrastive pretrained visual encoders.

Reviewer fT87 exhibits a neutral to slightly positive stance, valuing the insights on contrastive pretraining and the strong empirical results. However, the reviewer notes that the comparative analysis with only SigLIP models lacks representativeness. The authors agree on this point and promise to try to address in the final version with additional OpenCLIP baselines.

Reviewer Cwas expresses strong negative attitude on the acceptance of this paper. The reviewer acknowledges the strong empirical results but questioning the novelty of the findings. The reviewer challenges that the claim of contrastively trained backbones' superiority is an already recognized fact in the field. Another concern of the reviewer is that PaLI-3's advancements on training recipes is incremental compared with previous literature. The rebuttal failed to sway this opinion.

Despite the strong empirical results and extensive experiments by the authors, a unanimous concern arises from the limited scope of comparison experiments with SigLIP. This narrow comparison undermines the paper's central claim regarding the efficacy of contrastively pretrained visual backbones in visually-situated text understanding and localization tasks. Furthermore, as Reviewer Cwas and prior literature (e.g., [1]) indicate, the asserted superiority of contrastively pretrained backbones is not a novel concept in the field. Given that this aspect is pivotal to the paper's contributions, its novelty is significantly diminished.

Considering these factors, particularly the need for a more expansive and representative set of comparative experiments and the issue of novelty, the AC decides to reject this submission.

**Justification For Why Not Higher Score:**

Despite the strong empirical results and extensive experiments by the authors, a unanimous concern arises from the limited scope of comparison experiments with SigLIP. This narrow comparison undermines the paper's central claim regarding the efficacy of contrastively pretrained visual backbones in visually-situated text understanding and localization tasks. Furthermore, as Reviewer Cwas and prior literature (e.g., [1]) indicate, the asserted superiority of contrastively pretrained backbones is not a novel concept in the field. Given that this aspect is pivotal to the paper's contributions, its novelty is significantly diminished.

**Justification For Why Not Lower Score:**

N/A

---

### Decision · Program_Chairs · 2024-01-16

Reject